# PART-AWARE CLIP: ENHANCING FINE-GRAINED UNDERSTANDING WITH PART-LEVEL DESCRIPTIONS

## ABSTRACT

Vision-Language Pre-trained (VLP) models, such as CLIP, learn powerful representations from large-scale image-text pairs, achieving remarkable zero-shot capabilities. However, by learning to align visual and textual features only at a global level, these models exhibit two critical limitations: poor interpretability and weak fine-grained perception. This deficiency arises from the pre-training data that overlooks key visual details of object parts, which are often important for distinguishing between subordinate categories (e.g., species of birds or models of cars). This fundamental weakness is inherited by Multimodal Large Language Models (MLLMs) that use CLIP-based vision encoders, limiting both their accuracy and trustworthiness. To address these challenge, we introduce Part-Aware CLIP (*PA-CLIP*), a framework designed to enhance both the fine-grained perception and interpretability of VLP models. First, we employs MLLMs to create a new dataset *FG-Part*, including approximately 1 million part-level image-text pairs. FG-Part explicitly captures the critical visual details of object components (e.g., a bird's beak and wing patterns). Second, we design a part-aware training strategy that leverages this curated data to compel the model to ground fine-grained textual descriptions in specific image regions. By forcing this explicit, part-level alignment, our method enhances the model's ability to perceive key details, thereby improving its fine-grained recognition capabilities and inherent interpretability. Extensive experiments show that PA-CLIP achieves state-of-the-art performance on multiple fine-grained visual recognition (FGVR) benchmarks, highlighting the effectiveness of part-level captions and the capabilities of PA-CLIP in capturing subtle visual details. Furthermore, evaluations on general tasks, including cross-modal retrieval, confirm that these gains do not compromise the model's core generalist capabilities.

## 1 INTRODUCTION

The advent of large-scale pre-training has marked a paradigm shift in artificial intelligence, with Vision-Language Pre-trained (VLP) models leading the charge in multimodal understanding. Models such as CLIP (Contrastive Language-Image Pre-training) (Radford et al., 2021), SigLIP (Zhai et al., 2023) have demonstrated remarkable success by learning rich, transferable visual representations from hundreds of millions of image-text pairs from the web. Demonstrating robust zero-shot capabilities in visual understanding, these models serve as a powerful foundation for a wide range of downstream task, such as image classification (Radford et al., 2021), cross-modal retrieval (Csizmadia et al., 2025). Notably, in recent years, influential MLLMs such as the LLaVA (Liu et al., 2023; 2024) and MiniGPT (Zhu et al., 2023; Chen et al., 2023) series have widely adopted pre-trained CLIP models as their vision encoder to extract visual representations.

Despite their impressive capabilities, CLIP series models exhibit a critical weakness when dealing with tasks that demand a fine-grained understanding of visual information (Xie et al., 2025a; Asokan et al., 2025a). By learning to align images and text at a coarse, global level, CLIP and its successors often fail at fine-grained visual recognition (FGVR)—the task of distinguishing between subordinate categories, such as species of birds, models of cars, or types of aircraft (**?**Welinder et al., 2010; Maji et al., 2013a). This deficiency arises as the pre-training process overlooks the subtle, part-level visual details (e.g., the shape of a bird's beak, the curvature of a car's headlights) that are often the sole differentiators between fine-grained classes. Recent works on improving the fine-grained

understanding abilities of CLIP models has primarily concentrated on two main directions. The first approach (Zhang et al., 2024c; Xie et al., 2025a) focuses on extending the text token sequence beyond the 77-token limit of the original CLIP models. This allows the model to process longer and more detailed captions during the training phase, which contains more textual information. A second, more direct strategy involves fine-tuning the model on curated, fine-grained image-text datasets (Xie et al., 2025a; Xiao et al., 2024). This method provides explicit fine-grained supervision, teaching the model to associate specific visual features with their corresponding textual descriptions.

However, while these approaches have incrementally improved the fine-grained capabilities of CLIP, they still suffer from two significant drawbacks: a persistent lack of interpretability (Rasekh et al., 2024) and insufficient fine-grained understanding. On challenging FGVR benchmarks, their accuracy still lags considerably behind models designed specifically for these tasks. The primary reason for this persistent gap is that even the fine-grained image-text data remains fundamentally "coarse-grained." Although previous methods provide longer, more detailed text, these descriptions often focus on the holistic scene rather than the discriminative, part-level attributes. For instance, the critical visual distinction between two visually similar bird species, the 'Great Crested Flycatcher' and the 'White-browed Chat-tyrant', may lie solely in a subtle pattern on the wing feathers—a minute detail that is decisive for correct classification. Existing image-text datasets, even those curated for fine-grained tasks, are unable to consistently capture such subtle, part-level visual differences in their textual descriptions. Consequently, models trained on this data are not equipped with the necessary supervision to learn these critical features, leading to an inability to reliably discern between visually similar fine-grained categories.

To overcome these fundamental limitations, we introduce Part-Aware CLIP (PA-CLIP), a novel framework that systematically enhances both the fine-grained perception and interpretability of VLP models. First, we propose a scalable data generation pipeline which utilizes powerful MLLMs to generate fine-grained image-text data. This pipeline allows us to construct a large-scale dataset named FG-Part, which included approximately 1 million part-level image-text pairs. Unlike previous long textual captions, our generated descriptions function as explicit textual pointers to discriminative, part-level visual evidence (e.g., "The throat of the bird exhibits a pale bluish-gray color, with a relatively smooth texture"), capturing the kind of subtle features that are essential for expert-level recognition but absent in existing datasets.

Second, building upon this curated dataset, we design a part-aware fine-tuning strategy. This strategy explicitly compels the model to ground these nuanced textual descriptions in their corresponding local image regions, forcing an alignment between specific words and the visual features they describe. This process improves the model's understanding of part-level semantics, improving both its fine-grained recognition accuracy and its inherent interpretability, as the model's attention is now guided by discriminative object characteristics.

On several challenging FGVR benchmarks, our method consistently surpasses existing state-of-the-art approaches. Concurrently, experiments on general vision-language benchmarks, such as cross-modal retrieval, demonstrate that PA-CLIP maintains highly competitive performance with existing methods. Experimental results prove that our approach enhances fine-grained perception while preserving the versatile, generalist capabilities of the foundational CLIP architecture.

## 2 RELATED WORK

### 2.1 CONTRASTIVE VISION-LANGUAGE PRE-TRAINING

Learning visual representations from natural language supervision has become a dominant paradigm in computer vision. This approach was popularized by models like CLIP (Radford et al., 2021) and ALIGN (Jia et al., 2021), which pioneered the use of contrastive learning on a large scale. The resulting models learn robust and generalizable representations that enable impressive zero-shot performance on a wide array of classification tasks without direct supervision. Following this initial success, research has largely advanced in two main directions. One dominant trend has been to scale up the training data. The creation of massive public datasets, most notably the LAION series (Schuhmann et al., 2022), provided billions of image-text pairs. Building upon this larger, more diverse data enabled the development of more powerful open-source models like OpenCLIP (Cherti et al., 2023). A second direction has focused on improving the quality of the textual supervision.

Recognizing that noisy web captions are often short and lack detail, some works (Zhang et al., 2024b; Asokan et al., 2025b) have sought to improve CLIP's fine-grained understanding by using longer and more detailed text. This is often done by employing sophisticated captioning models to generate longer, more detailed sentences for the entire image or regions within images, with the goal of providing a richer and more textual information.

## 2.2 Fine-grained Visual Recognition

While recognizing general objects like "car" or "bird" is a largely solved problem, the next frontier is fine-grained understanding: the ability to perceive and reason about subtle, detailed visual information. This capability is essential for real-world applications, from identifying specific plant diseases in agriculture to describing nuanced product details in e-commerce. As discussed, the VLP models that form the basis of today's MLLMs lack this crucial skill due to their global-level pre-training. Fine-grained visual recognition (FGVR) is a essential for fine-grained understanding, as the model needs to accurately identify the fine-grained categories of objects before performing fine-grained analysis (e.g., visual question answering). Historically, research in FGVR has followed two main paths. The first involved specialized models (Zheng et al., 2019; Ji et al., 2020; Sun et al., 2023a) that were explicitly designed to perform fine-grained image classification. Based on CNN or transformers architecture, these methods show excellent performance on specific datasets. The second path involves using large Vision-Language Models for classification. The standard approach is to perform zero-shot classification by crafting a set of text prompts, such as "a photo of a class name" (Radford et al., 2021). The model then classifies an image by comparing its embedding to the embeddings of the different prompts.

## 3 Methodology

In this section, we introduce our proposed Part-Aware CLIP (PA-CLIP), a novel framework designed to inject fine-grained perception and interpretability into existing VLP models. Our approach tackles the inherent limitations of models like CLIP by explicitly grounding textual descriptions of object parts to their corresponding visual regions. This is achieved through a multi-level contrastive learning objective powered by a meticulously constructed dataset of part-level image-text pairs. The overall architecture of PA-CLIP is depicted in Figure 1.

## 3.1 PA-CLIP

Following previous work (Xie et al., 2025b; Xiao et al., 2024), we first extend the limitation of text token sequence in CLIP models from 77 to 248, enabling models to capture more detailed textual information. Our PA-CLIP framework adopts the same architecture as the original CLIP, comprising a text encoder and an image encoder, and introduces a specialized fine-tuning stage. This stage is orchestrated by a multi-level contrastive learning objective, which operates on both global image-text pairs and fine-grained, part-level correspondences. This dual-level supervision ensures that the model not only retains its robust general-purpose knowledge but also develops a deep understanding of subtle visual details critical for fine-grained tasks.

### 3.1.1 Image and Text Encoders

We adopt the standard dual-encoder architecture for our PA-CLIP, comprising an image encoder $f_I(\cdot)$ and a text encoder $f_T(\cdot)$. For the image encoder, we primarily utilize a Vision Transformer (ViT) (Dosovitskiy et al., 2020) architecture. When processing a full image $x$, the encoder $f_I(x)$ produces a global embedding $\mathbf{v}_g \in \mathbb{R}^D$ from its output [CLS] token, capturing the holistic content of the scene. The text encoder $f_T(\cdot)$ processes textual inputs (short captions, long captions, part descriptions) to produce corresponding $D$-dimensional text embeddings $\mathbf{t} \in \mathbb{R}^D$. All embeddings are L2-normalized before computing similarities.

### 3.1.2 Multi-Level Contrastive Loss

Our training objective for PA-CLIP is designed to simultaneously learn global image-text alignments and fine-grained, part-level correspondences. This is achieved by combining distinct contrastive loss

components. For a given mini-batch of $N$ image-text sets, the total loss $\mathcal{L}$ is defined as:

$$\mathcal{L} = \mathcal{L}_{\text{global}} + \alpha \cdot \mathcal{L}_{\text{part}} \tag{1}$$

where $\mathcal{L}_{\text{global}}$ is the global image-text contrastive loss, $\mathcal{L}_{\text{part}}$ is the part-level contrastive loss, and $\alpha$ is a tunable hyperparameter.

**Global Contrastive Loss**. To ensure PA-CLIP retains its broad zero-shot capabilities, we incorporate a global contrastive loss that considers two levels of global textual descriptions: short captions and long captions. For each image $x_i$ in the batch, we construct a short caption $y_{i,s}$ and a long caption $y_{i,l}$. Then, the corresponding text embeddings are $\mathbf{t}_{i,s} = f_T(y_{i,s})$ and $\mathbf{t}_{i,l} = f_T(y_{i,l})$. The global image embedding is $\mathbf{v}_{i,g} = f_I(x_i)_{\text{CLS}}$.

The global contrastive loss can be then expressed as:

$$\mathcal{L}_{\text{global}} = \mathcal{L}_{\text{CLIP}}(\mathbf{v}_g, \mathbf{t}_s) + \mathcal{L}_{\text{CLIP}}(\mathbf{v}_g, \mathbf{t}_l) \tag{2}$$

where $\mathcal{L}_{\text{CLIP}}(\mathbf{V}, \mathbf{T})$ is a standard symmetric contrastive loss function. Given a batch of $N$ image embeddings $\mathbf{V} = \{\mathbf{v}_1, \ldots, \mathbf{v}_N\}$ and $N$ text embeddings $\mathbf{T} = \{\mathbf{t}_1, \ldots, \mathbf{t}_N\}$, this loss is defined as:

$$\mathcal{L}_{\text{CLIP}}(\mathbf{V}, \mathbf{T}) = \frac{1}{2N} \sum_{i=1}^{N} \left[ -\log \frac{\exp(\text{sim}(\mathbf{v}_i, \mathbf{t}_i)/\tau)}{\sum_{j=1}^{N} \exp(\text{sim}(\mathbf{v}_i, \mathbf{t}_j)/\tau)} - \log \frac{\exp(\text{sim}(\mathbf{t}_i, \mathbf{v}_i)/\tau)}{\sum_{j=1}^{N} \exp(\text{sim}(\mathbf{t}_i, \mathbf{v}_j)/\tau)} \right] \tag{3}$$

Here, $\text{sim}(\cdot, \cdot)$ denotes cosine similarity, and $\tau$ is a learnable temperature parameter.

**Part-Level Contrastive Loss**. While the global loss ensures the model understands the overall scene, the part-level loss is specifically designed to focus on the fine-grained visual details within it. This component enforces a direct alignment between nuanced textual descriptions and their specific visual counterparts at the sub-object level. To achieve this, for each image $x_i$ annotated with $K_i$ object parts, we generate a set of dedicated part-specific visual embeddings.

Specifically, for each part $k$ of image $x_i$, defined by its bounding box $\text{BBox}_{i,p}^k$, we first crop this region from the original high-resolution image, yielding a part-specific image patch $x_{i,p}^k$. This patch is then resized to the standard input resolution of the image encoder and processed in a separate forward pass. The CLS token embedding from this pass serves as the dedicated visual feature for that part:

$$\mathbf{v}_{i,p}^k = f_I(x_{i,p}^k)_{\text{CLS}} \tag{4}$$

This approach yields a feature vector $\mathbf{v}_{i,p}^k$ that represents the part in isolation, which is crucial for learning subtle, context-independent visual attributes. The corresponding part-level text description $y_{i,p}^k$ is then encoded by the text encoder to produce its embedding $\mathbf{t}_{i,p}^k = f_T(y_{i,p}^k)$.

The part-level contrastive loss $\mathcal{L}_{\text{part}}$ is then computed by encouraging each part's visual feature $\mathbf{v}_{i,p}^k$ to be close to its matching textual description $\mathbf{t}_{i,p}^k$, and distant from all other part descriptions in the batch. This is a symmetric contrastive loss operating on part-level visual and textual embeddings:

$$\mathcal{L}_{\text{part}} = \mathcal{L}_{\text{CLIP}}(\mathbf{V}_P, \mathbf{T}_P) \tag{5}$$

where $\mathbf{V}_P$ and $\mathbf{T}_P$ represent the concatenated sets of all part visual embeddings and part text embeddings, respectively, within the current mini-batch. By forcing this explicit alignment between cropped part images and their detailed descriptions, PA-CLIP learns to perceive and ground discriminative local features in a way that global-only training cannot achieve.

### 3.2 FG-PART DATASET

The core challenge in instilling part-level awareness in VLP models is the scarcity of datasets containing explicit, high-quality descriptions of object parts. However, manually annotating hundreds of thousands of image parts with detailed textual descriptions is infeasible. To circumvent this bottleneck and enable scalable training, we developed a sophisticated, automated pipeline for generating a large-scale dataset named FG-Part of part-level image-text pairs using MLLMs.

**Part Localization and Extraction.** First, we detect the discriminative regions of objects within an image. Specifically:

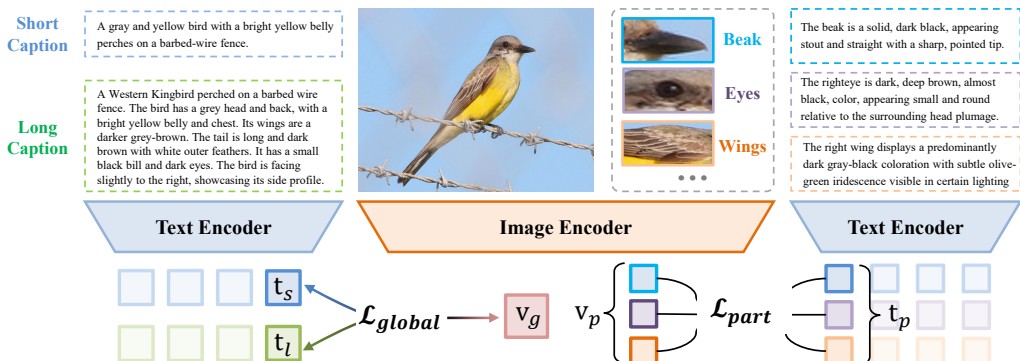

Figure 1: An overview of our proposed PA-CLIP framework. The global loss ($\mathcal{L}_{\text{global}}$) aligns the global image representation ($\mathbf{v}_g$) with embeddings from both short ($\mathbf{t}_s$) and long ($\mathbf{t}_l$) captions. Concurrently, the part-level loss ($\mathcal{L}_{\text{part}}$) aligns dedicated part-level visual features ($\mathbf{v}_p$) with their corresponding textual embeddings ($\mathbf{t}_p$).

- **For datasets with existing part annotations:** For established fine-grained datasets such as CUB-200-2011 (Wah) and Birdsnap (Berg et al., 2014), which already provide high-quality human-annotated part-level bounding boxes (e.g., 'beak', 'wing', 'eyes' for birds), we directly utilize these ground-truth annotations to extract object parts.

- **For datasets without part annotations:** To vastly expand our data sources to datasets that lack part labels (e.g., StanfordCars (Krause et al., 2013a) and FGVC-Aircraft (Maji et al., 2013b)), we employ an open-vocabulary object detection model, OV-DINO (Wang et al., 2024a). We first curate a domain-specific *part name list* containing key discriminative components (e.g., for cars: 'logo', 'wheel hub', 'headlight', 'grille'). We then use this list to query the OV-DINO model, which automatically localizes these predefined parts within the images and generates the corresponding bounding boxes.

For each obtained bounding box, we crop the corresponding image region, creating a collection of individual part images for the next stage.

**Prompt Design for Description Generation.** To elicit high-quality descriptions from the MLLM, we carefully designed prompt templates for different objects. A prompt template is as follows:

```
Object SuperClass:  Bird
Object Attributes:  Beak, Wings, Belly, Crown
Examples:
- #1 Harris Sparrow's tail is moderately long and slightly...
- #2 This Common Tern's belly displays a predominantly white
coloration, exhibiting a soft, fluffy texture...
- #3 The Red-headed Woodpecker's crown is a vibrant, red...
Instruction:  You are a professional ornithologist and computer
vision expert.  Please generate detailed textual descriptions
for the given bird image regions.
```

Specifically, we utilized Google's Gemma3 27B (Gemma Team, 2025) and Meta's Llama4 Scout

(Meta AI, 2025) for image understanding and description generation. These models were deployed across 16x NVIDIA 5090 and 20x NVIDIA L40 GPUs using vllm toolkit (Kwon et al., 2023).

Through this pipeline, we curated our FG-Part dataset of approximately 1 million part-level image-text pairs. This dataset provides fine-grained supervision, addressing the absence of localized detail in standard VLP training corpora.

## 4 EXPERIMENTS

### 4.1 EXPERIMENTAL SETTINGS

**Training.** Following previous work (Xie et al., 2025b), we utilize FineHARD (Xie et al., 2025b) and the curated FG-Part dataset for training. FineHARD dataset contains over 12 million images along with fine-grained descriptions, 40 million bounding boxes with corresponding captions and 10 million hard negative examples.

**Implementation Details.** Following Xie et al. (2025b); Asokan et al. (2025b), we initialze our models using the original CLIP weights (Radford et al., 2021). Our experiments are conducted on both CLIP-ViT-Base/16 (ViT-B-16) and CLIP-ViT-Large/14 (ViT-L-14) configurations. For the first stage of training on the FineHARD dataset, we utilize a cluster of 10 NVIDIA A6000 GPUs. We set the per-GPU batch size to 128, resulting in a global batch size of 1280. We employ the AdamW optimizer with a learning rate of 1e-6, and utilize a cosine annealing learning rate scheduler. The learnable temperature parameter, $\tau$, in the contrastive loss is initialized to 0.07. We leverage the DeepSpeed ZeRO stage 2 optimization configuration to accelerate the training process. In the second stage, we train the model on our FG-Part dataset using a setup of 8 NVIDIA RTX 5090 GPUs. The per-GPU batch size is set to 64, yielding a global batch size of 512. We use the AdamW optimizer with a learning rate of 1e-6 and a cosine annealing schedule. We set a warm-up period of 300 steps and train for a total of 20 epochs. The DeepSpeed ZeRO stage 2 configuration is employed to optimize the training process.

**Evaluation Tasks and Metrics.** We conduct extensive experiments across two tasks: Fine-Grained Visual Recognition (FGVR) and cross-modal retrieval. FGVR is a core challenge that directly measures a model's ability to distinguish between highly similar subordinate categories. We use the Top-1 Accuracy as the evaluation metric, which measures the percentage of test images correctly classified into their ground-truth category. Cross-Modal Retrieval Wang et al. (2024b) evaluates the model's general-purpose representation quality for aligning images and text. We report performance on both text-to-image retrieval (T2I) and image-to-text retrieval (I2T) setting. The key metric is Recall@1 (R@1), which measures the percentage of queries where the correct corresponding item is ranked as the top result.

**Evaluation Methods.** Baselines include the original CLIP (Radford et al., 2021), EVA-CLIP (Sun et al., 2023b), SigLIP (Zhai et al., 2023), DreamLIP (Zheng et al., 2024), and recent fine-grained focused models like Long-CLIP (Zhang et al., 2024c) and FG-CLIP (Xie et al., 2025b), which extends the text length limitation. Results are reported for both ViT-B-16 and ViT-L-14 configurations.

### 4.2 RESULTS

We conduct a comprehensive evaluation of PA-CLIP to demonstrate its capabilities in both specialized and general domains. The primary evaluation of our method's fine-grained perception is conducted on several challenging FGVR benchmarks. Furthermore, to ensure that our improvements in fine-grained perception do not come at the cost of generalist capabilities, we also evaluate out method on a diverse set of cross-modal retrieval tasks.

#### 4.2.1 FINE-GRAINED VISUAL RECOGNITION.

For FGVR task, we ultize a wide range of standard FGVR datasets, covering diverse domains from birds to cars to general scenes. These include CUB-200-2011 (CUB) (Wah), FGVC Aircraft (Aircraft) (Maji et al., 2013b), Stanford Dogs (Dogs) (Khosla et al., 2011), Oxford-IIIT Pets (Pets) (Parkhi et al., 2012), SUN397 (Sun) (Xiao et al., 2010), NABirds (Van Horn et al., 2015), Oxford Flowers (Flowers) (Nilsback & Zisserman, 2008), and Stanford Cars (Cars) (Krause et al., 2013b).

The main results for fine-grained visual recognition are presented in Table 1. Our proposed PA-CLIP demonstrates a substantial and consistent improvement over all baseline and state-of-the-art methods across both ViT-B-16 and ViT-L-14 backbones. Specifically, with the ViT-L-14 backbone, PA-CLIP achieves the state-of-the-art average accuracy of 79.3% across the eight FGVR datasets. This result surpasses the next best competing method, SigCLIP, by 8.4%. With the ViT-B-16 backbone, PA-CLIP achieves an average accuracy of 71.1%, outperforming the strongest baseline (SigCLIP at

Table 1: Comparison of FGVR performance on 8 benchmarks. Top-1 accuracy is reported.

| Method | Backbone | CUB | Aircraft | Dogs | Pets | Sun | Nabirds | Flowers | Cars | Average |
|---|---|---|---|---|---|---|---|---|---|---|
| CLIP | ViT-B/16 | 48.1 | 23.4 | 57.1 | 84.4 | 64.9 | 42.4 | 64.5 | 61.7 | 55.8 |
| EVA-CLIP | ViT-B/16 | 51.7 | 20.9 | 63.4 | 86.8 | 68.3 | 49.1 | 70.4 | 79.5 | 61.3 |
| Long-CLIP | ViT-B/16 | 46.4 | 21.6 | 54.3 | 83.1 | 62.6 | 40.1 | 62.8 | 61.0 | 54.0 |
| SigCLIP | ViT-B/16 | 38.1 | 36.7 | 66.4 | 87.5 | 66.6 | 50.5 | 83.9 | **91.1** | 65.1 |
| DreamLIP | ViT-B/16 | 33.0 | 7.9 | 22.8 | 64.1 | 63.9 | 27.2 | 51.8 | 22.7 | 36.7 |
| FG-CLIP | ViT-B/16 | 55.0 | 22.6 | 61.8 | 88.0 | **70.2** | 48.3 | 68.0 | 85.9 | 62.5 |
| PA-CLIP | ViT-B/16 | **68.8** | **36.9** | **72.6** | **93.8** | 68.1 | **53.4** | **87.8** | 87.4 | **71.1** |
| CLIP | ViT-L/14 | 56.1 | 32.0 | 66.4 | 89.8 | 69.6 | 51.5 | 72.0 | 73.0 | 63.8 |
| EVA-CLIP | ViT-L/14 | 61.5 | 32.6 | 69.2 | 87.5 | 72.0 | 57.6 | 75.3 | 90.2 | 68.2 |
| Long-CLIP | ViT-L/14 | 45.1 | 27.9 | 56.2 | 82.7 | 65.8 | 47.7 | 63.2 | 71.5 | 57.5 |
| SigCLIP | ViT-L/14 | 52.5 | 43.4 | 74.0 | 92.7 | 69.3 | 58.6 | 84.6 | 92.4 | 70.9 |
| FG-CLIP | ViT-L/14 | 64.2 | 29.8 | 69.9 | 90.4 | 74.2 | 59.9 | 70.7 | 90.0 | 68.6 |
| PA-CLIP | ViT-L/14 | **78.6** | **52.7** | **80.6** | **96.5** | **74.3** | **66.4** | **92.6** | **92.7** | **79.3** |

65.1%) by 6.0%. This consistent superiority across different model scales highlights the effectiveness of our PA-CLIP and FG-Part dataset.

On individual datasets, PA-CLIP sets new SOTA results on all benchmarks with the ViT-L/14 backbone. The gains are particularly significant on datasets where localized features are critical for recognition. For instance, on Aircraft, PA-CLIP-L-14 achieves 52.7% accuracy, outperforming the CLIP-L-14 baseline by +20.7%. On CUB and Dogs, we observe improvements of 22.5% and 14.2%, respectively. This demonstrates that our method is highly effective at capturing the discriminative, part-level features required for challenging fine-grained tasks. Conversely, on broad, scene-level datasets like SUN397, where global context is more dominant than specific object parts, the performance improvement of PA-CLIP is relatively modest.

Table 2: Comparison of cross-modal retrieval performance on 5 benchmarks. Recall@1 (R@1) is reported. I2T and T2I indicate Image-to-Text retrieval and Text-to-Image retrieval respectively.

| Method | Backbone | MSCOCO | | DCI | | Flickr | | DOCCI | | Urban1k | |
|---|---|---|---|---|---|---|---|---|---|---|---|
| | | I2T | T2I | I2T | T2I | I2T | T2I | I2T | T2I | I2T | T2I |
| CLIP | ViT-B/16 | 51.8 | 32.7 | 45.5 | 43.0 | 82.2 | 62.1 | - | - | - | - |
| EVA-CLIP | ViT-B/16 | 58.7 | 41.6 | 41.9 | 41.2 | 85.7 | 71.2 | - | - | - | - |
| Long-CLIP | ViT-B/16 | 57.6 | 40.4 | 51.7 | 57.3 | 85.9 | 70.7 | - | - | 78.9 | 79.5 |
| FineCLIP | ViT-B/16 | 54.5 | 40.2 | 35.5 | 34.4 | 82.5 | 67.9 | 78.1 | 80.0 | 91.2 | 90.0 |
| FG-CLIP | ViT-B/16 | 64.1 | 45.4 | **61.8** | **60.6** | **90.9** | 75.5 | **83.9** | **84.4** | **94.0** | **93.4** |
| PA-CLIP | ViT-B/16 | **65.3** | **46.3** | 60.8 | 59.2 | 88.9 | **76.1** | 83.5 | 83.3 | 93.9 | 92.9 |
| CLIP | ViT-L/14 | 58.0 | 37.1 | 37.2 | 36.4 | 87.4 | 67.3 | - | - | - | - |
| EVA-CLIP | ViT-L/14 | 64.2 | 47.9 | 47.2 | 47.8 | 89.2 | 77.9 | - | - | - | - |
| Long-CLIP | ViT-L/14 | 62.8 | 46.3 | 44.2 | 52.5 | 90.0 | 76.2 | - | - | 82.7 | 86.1 |
| FineCLIP | ViT-L/14 | - | - | 40.1 | 46.2 | - | - | 83.7 | 84.4 | 94.5 | 93.9 |
| FG-CLIP | ViT-L/14 | 68.9 | 50.9 | **66.7** | **66.1** | 93.7 | 81.5 | **87.4** | **89.3** | 96.4 | 96.5 |
| PA-CLIP | ViT-L/14 | **70.3** | **51.4** | 65.6 | 65.6 | **94.1** | **81.9** | 87.3 | 87.8 | **97.1** | **96.7** |

### 4.2.2 CROSS-MODAL RETRIEVAL

To verify that PA-CLIP maintains robust general-purpose representations, we evaluate its performance on standard cross-modal retrieval benchmarks. We select five datasets to test retrieval performance across different domains and different caption length. These include the two canonical benchmarks, MSCOCO 5K (MSCOCO) Lin et al. (2014), a subset of MSCOCO dataset with 5000 images of complex everyday scenes, and Flickr 1K (Flickr) (Young et al., 2014), with 1000 images of everyday activities, both with five captions per image. To test understanding of more complex and longer text, we also evaluate on DOCCI (Onoe et al., 2024), a challenging benchmark with 5000

images for evaluation; DCI (Urbanek et al., 2024), with 7,805 image-text pairs and Urban1K (Zhang et al., 2024a), including 1000 images with corresponding long captions.

The results for cross-modal retrieval are presented in Table 2. The primary goal of this evaluation is to verify that PA-CLIP's enhanced fine-grained perception does not degrade its general-purpose representation quality. Experimental results indicate that PA-CLIP achieves comparable performance with SOTA methods.

Table 3: Ablation studies on fine-grained visual recognition. We report the average Top-1 accuracy (%) across 8 FGVR benchmarks for different combinations of long global (#*Long*), short global (#*Short*), and part-level (#*Part*) captions.

| #*Long* | #*Short* | #*Part* | Backbone | CUB | Aircraft | Dogs | Pets | Sun | Nabirds | Flowers | Cars | Average |
|---|---|---|---|---|---|---|---|---|---|---|---|---|
| ✓ | | | ViT-B-16 | 59.9 | 25.7 | 61.2 | 88.7 | 69.1 | 53.3 | 67.6 | 83.9 | **63.7** |
| | ✓ | | ViT-B-16 | 60.3 | 36.8 | 66.5 | 89.7 | 69.2 | 54.4 | 69.0 | 87.4 | **66.7** |
| | | ✓ | ViT-B-16 | 65.2 | 30.7 | 70.7 | 93.9 | 67.8 | 49.2 | 84.8 | 86.9 | **68.6** |
| | ✓ | ✓ | ViT-B-16 | 68.3 | 34.3 | 71.4 | 93.6 | 67.9 | 53.7 | 87.1 | 87.5 | **70.5** |
| ✓ | | ✓ | ViT-B-16 | 68.4 | 29.8 | 70.6 | 93.9 | 67.6 | 53.3 | 86.5 | 86.1 | **69.5** |
| ✓ | ✓ | | ViT-B-16 | 60.2 | 36.9 | 66.0 | 89.0 | 69.2 | 54.0 | 69.1 | 87.2 | **66.5** |
| ✓ | ✓ | ✓ | ViT-B-16 | 68.8 | 36.9 | 72.6 | 93.8 | 68.1 | 53.4 | 87.8 | 87.4 | **71.1** |
| ✓ | | | ViT-L-14 | 70.5 | 35.7 | 70.0 | 88.7 | 73.6 | 63.8 | 71.6 | 90.1 | **70.5** |
| | ✓ | | ViT-L-14 | 70.9 | 50.8 | 73.4 | 89.6 | 73.8 | 66.3 | 73.3 | 91.9 | **73.7** |
| | | ✓ | ViT-L-14 | 77.6 | 43.2 | 78.9 | 95.8 | 73.2 | 65.3 | 89.4 | 92.1 | **76.9** |
| | ✓ | ✓ | ViT-L-14 | 78.2 | 50.8 | 79.5 | 94.8 | 73.6 | 67.3 | 90.9 | 92.3 | **78.4** |
| ✓ | | ✓ | ViT-L-14 | 77.9 | 40.4 | 78.2 | 95.4 | 73.2 | 66.3 | 90.7 | 92.0 | **76.8** |
| ✓ | ✓ | | ViT-L-14 | 70.3 | 49.1 | 72.6 | 89.6 | 73.9 | 65.1 | 72.7 | 91.8 | **73.1** |
| ✓ | ✓ | ✓ | ViT-L-14 | 78.6 | 52.7 | 80.6 | 96.5 | 74.3 | 66.4 | 92.6 | 92.7 | **79.3** |

On the canonical benchmarks, MSCOCO and Flickr, PA-CLIP consistently sets new state-of-the-art results for the ViT-L-14 backbone. Specifically, on MSCOCO, our PA-CLIP-L-14 achieves 70.3% on Image-to-Text (I2T) retrieval and 51.4% on Text-to-Image (T2I) retrieval, outperforming the FG-CLIP. A similar trend is observed on Flickr, where our model also achieves top performance. This demonstrates that our part-aware description is compatible with and can even enhance performance on general-domain retrieval. On datasets characterized by long and complex captions, such as DOCCI, DCI, and Urban1k, PA-CLIP exhibits highly competitive performance compared with other methods. For example, with the ViT-L-14 bakcbone, PA-CLIP achieves a SOTA result of 97.1% on Urban1k I2T, while being within 0.1% of the best score on DOCCI I2T. The results indicate that our model does not lose its ability to understand long text while improving fine-grained perception capability.

Table 4: Ablation studies on cross-modal retrieval. We report the average Recall@1 (R@1) across 5 benchmarks for different combinations of long global (#*Long*), short global (#*Short*), and part-level (#*Part*) captions.

| *Long* | *Short* | *Part* | Backbone | MSCOCO | | DCI | | Flickr | | DOCCI | | Urban1k | | Average |
|---|---|---|---|---|---|---|---|---|---|---|---|---|---|---|
| | | | | I2T | T2I | I2T | T2I | I2T | T2I | I2T | T2I | I2T | T2I | |
| ✓ | | | ViT-B-16 | 66.2 | 46.3 | 62.2 | 60.1 | 91.0 | 75.9 | 84.7 | 85.2 | 93.2 | 93.2 | **75.8** |
| | ✓ | | ViT-B-16 | 64.9 | 47.6 | 60.9 | 61.3 | 89.5 | 76.9 | 84.0 | 85.2 | 92.9 | 92.3 | **75.5** |
| | | ✓ | ViT-B-16 | 62.2 | 41.9 | 57.3 | 53.2 | 89.2 | 73.5 | 78.0 | 77.4 | 90.7 | 89.0 | **71.2** |
| | ✓ | ✓ | ViT-B-16 | 63.9 | 44.8 | 57.6 | 56.8 | 88.8 | 75.1 | 81.3 | 81.2 | 92.7 | 91.2 | **73.3** |
| ✓ | | ✓ | ViT-B-16 | 64.5 | 43.4 | 59.7 | 56.1 | 90.1 | 74.4 | 82.5 | 81.5 | 93.6 | 92.5 | **73.8** |
| ✓ | ✓ | | ViT-B-16 | 65.4 | 47.3 | 62.4 | 60.3 | 89.8 | 76.6 | 84.6 | 85.0 | 92.9 | 92.4 | **75.7** |
| ✓ | ✓ | ✓ | ViT-B-16 | 65.3 | 46.3 | 60.8 | 59.2 | 88.9 | 76.1 | 83.5 | 83.3 | 93.9 | 92.9 | **75.0** |
| ✓ | | | ViT-L-14 | 71.3 | 51.9 | 68.6 | 67.5 | 94.5 | 81.9 | 88.7 | 90.1 | 97.4 | 97.6 | **81.0** |
| | ✓ | | ViT-L-14 | 70.8 | 53.4 | 65.8 | 67.8 | 94.0 | 83.0 | 87.8 | 90.4 | 96.8 | 96.8 | **80.7** |
| | | ✓ | ViT-L-14 | 68.1 | 49.4 | 62.4 | 63.5 | 92.8 | 79.8 | 83.5 | 84.6 | 95.4 | 95.3 | **77.5** |
| | ✓ | ✓ | ViT-L-14 | 69.6 | 51.0 | 64.3 | 65.4 | 94.0 | 81.5 | 86.0 | 87.1 | 96.6 | 96.1 | **79.2** |
| ✓ | | ✓ | ViT-L-14 | 70.4 | 50.1 | 66.0 | 64.4 | 94.7 | 80.7 | 86.4 | 86.9 | 96.6 | 96.7 | **79.3** |
| ✓ | ✓ | | ViT-L-14 | 71.1 | 53.3 | 68.1 | 67.7 | 94.2 | 82.8 | 88.8 | 90.3 | 97.3 | 97.9 | **81.1** |
| ✓ | ✓ | ✓ | ViT-L-14 | 70.3 | 51.4 | 65.6 | 65.6 | 94.1 | 81.9 | 87.3 | 87.8 | 97.1 | 96.7 | **79.8** |

## 4.3 ABLATION STUDIES

We conduct a detailed ablation study to validate the effectiveness PA-CLIP. We report results on both FGVR (Table 3) and cross-modal retrieval (Table 4) tasks. In our experiments, we train the model with different caption types: long captions (#*Long*), short captions (#*Short*), and part-level captions (#*Part*).

**Fine-Grained Visual Recognition.** The results in Table 3 show that our part-level supervision is the most critical component for FGVR performance. Training with part-level supervision alone yields an average accuracy of 76.9% on the ViT-L-14 model, substantially outperforming models trained only on long (70.5%) or short (73.7%) global captions. Combining short and part supervision further improves performance to 78.4%, indicating a positive synergy between global context and local fine-grained details. Our full model, utilizing all three components, achieves the highest average accuracy of 79.3%, demonstrating that while part-level data is the primary driver, incorporating varied global supervision provides additional benefits.

**Cross-Modal Retrieval.** The results of ablation studies on cross-modal retrieval are summarized in Table 4. The average performance is highest when training exclusively on long or short global captions (#*Long* or #*Short*). Conversely, using only part-level supervision (#*Part*) leads to a notable performance degradation. This is expected, as part-level descriptions lack the global scene context essential for cross-modal retrieval. For instance, with the ViT-B-16 backbone, the Part-only model's average R@1 drops by 4.6% compared to the Long-only variant (71.2% vs. 75.8%). Crucially, our full model almost mitigates this gap, achieving a competitive performance of 75.0%. This demonstrates that combining global and part-level supervision effectively preserves general capabilities.

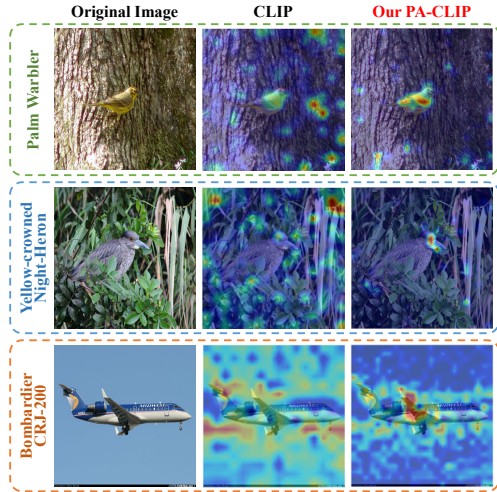

**Visualization.** To qualitatively assess how PA-CLIP enhances fine-grained perception ability, we visualize the vision encoder's attention maps using Grad-CAM (Selvaraju et al., 2019), shown in Figure 2. The visualizations reveal that the baseline CLIP model's attention is often diffuse, focusing on irrelevant background context such as tree bark or foliage. In contrast, PA-CLIP consistently and precisely localizes the object of interest. For instance, for the *Palm Warbler*, CLIP incorrectly focuses on the surrounding tree bark, while PA-CLIP focuses on the wings and tail region. For the *Yellow-crowned Night-Heron*, PA-CLIP's attention is tightly focused on the bird's discriminative head and eye region, rather than being scattered across the background. This ability to focus on discriminative object parts provides qualitative evidence for PA-CLIP's improved fine-grained performance.

Figure 2: Visualization of PA-CLIP.

## 5 CONCLUSION

In this paper, we addressed the critical limitations of standard Vision-Language Pre-trained (VLP) models: their poor fine-grained perception and lack of interpretability, which stem from a reliance on global-only image-text supervision. We introduced Part-Aware CLIP (PA-CLIP), a framework which introduces a multi-level contrastive loss to instill part-level awareness into these models. Extensive experiments demonstrate that PA-CLIP achieves new state-of-the-art performance on a comprehensive suite of fine-grained visual recognition benchmarks. In addition, we introduce FG-Part, a novel dataset of approximately 1 million image-text pairs with part-level, fine-grained descriptions.

**Future Work.** For future work, we identify two primary directions. First, we plan to further scale the FG-Part dataset, expanding its coverage to a wider diversity of domains and object categories. Second, a key avenue is the integration of PA-CLIP as an improved vision encoder for Multimodal Large Language Models (MLLMs) to enhance their fine-grained perceptual capabilities.

# 6 REPRODUCIBILITY STATEMENT

The MLLMs employed to construct the FG-Part dataset are publicly available, with detailed information of the specific models and the GPU server deployment provided in Section 3.2. All baseline methods and datasets used in our comparison experiments are also publicly available, with specifics provided in Section 4.1. All hyperparameters and training configurations required to reproduce our main results are detailed in Section 4.1.

# 7 STATEMENT ON THE USE OF LARGE LANGUAGE MODELS

During the writing process of this paper, LLMs were used only for language polishing and error correction. We affirm that all suggestions from these models have been strictly reviewed by the authors.

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
