# OpenReview forum: "Part-Aware CLIP: Enhancing Fine-Grained Understanding with Part-level Descriptions"
_ICLR.cc/2026/Conference — Submitted to ICLR 2026_

### Official Review · Reviewer_81Kv · 2025-10-29

**Soundness:** 2
**Presentation:** 3
**Contribution:** 2
**Rating:** 2
**Confidence:** 4

**Summary:**

This paper introduces Part-Aware CLIP (PA-CLIP), a framework addressing VLP models’ poor fine-grained perception and interpretability by leveraging part-level image-text alignment. It constructs the FG-Part dataset via MLLMs and uses multi-level contrastive learning, achieving SOTA on FGVR benchmarks while preserving general cross-modal retrieval capabilities

**Strengths:**

1. The paper is well-written and easy to follow.
2. The proposed method can effectively enhance the model's ability to perceive parts.

**Weaknesses:**

1. The proposed method has limited innovation and is more like a simplified version of FG-CLIP. Additionally, it lacks comparisons with relevant methods (e.g., [1], [2]).
2. The paper only verifies the model's performance on two tasks, namely Fine-Grained Visual Recognition (FGVR) and cross-modal retrieval, while other comparative experiments with related works (FG-CLIP, [1], [2]) are not included.
3. The results in Tables 3 and 4 indicate that the three different loss components lead to a trade-off between the model's global capability and part perception capability, which fails to support the claim in lines 31-33.

[1] UMG-CLIP: A Unified Multi-Granularity Vision Generalist for Open-World Understanding. ECCV 2024.

[2] Contrastive Localized Language-Image Pre-Training. ICML 2025.

**Questions:**

Should "SigCLIP" in Table 1 be "SigLIP"?

---

### Official Review · Reviewer_VxCK · 2025-10-31

**Soundness:** 4
**Presentation:** 3
**Contribution:** 2
**Rating:** 4
**Confidence:** 5

**Summary:**

PA-CLIP aims to address the limitations of VLP models in fine-grained recognition. The authors attribute this limitation to a lack of part-level supervision and propose two main contributions: 1) Constructing a large-scale (approx. 1M) part-level image-text dataset (FG-Part); 2) Designing a part-aware training strategy. This strategy maintains global alignment while introducing a contrastive loss between part-image patches and their detailed descriptions. Experimental results show that the method achieves SOTA on multiple FGVR benchmarks without sacrificing general cross-modal retrieval capabilities.

**Strengths:**

1. FG-Part innovatively constructs "Part-Aware descriptions" rather than conventional "Region-text captions," which contributes to the interpretable, fine-grained capabilities of CLIP models. The paper would be more persuasive if the FG-Part pipeline could be extended to general datasets and its effectiveness validated.
2. The main experiments on the FGVR task show excellent performance, and the ablation studies are thorough, effectively validating the contribution of the FG-Part data.

**Weaknesses:**

1. According to the FG-CLIP paper, the FineHard dataset is constructed from GRIT (from Coyo) and does not include data from CUB, Aircraft…, which are used for the FGVR evaluation. However, according to lines 234-245 of this paper, the FG-Part data is extracted from CUB, Aircraft, Dogs, etc., rather than from general datasets. This means the pre-training directly embeds prior knowledge of the "subordinate categories" (e.g., species of birds or models of cars) being evaluated. Consequently, the performance gain from FG-Part is intuitive and predictable, rather than revealing a novel finding.
2. While the zero-shot classification setting used is not directly comparable to prompt learning or linear probing (which achieve very high performance after fine-tuning on these datasets), I still recommend the authors design at least one experiment that uses the FG-Part data for prompt engineering. This could help demonstrate the necessity of the proposed pre-training stage.
3. Lack of comparison and references: The paper fails to compare with or cite other relevant methods that also use region-text data for contrastive pretraining, such as UMG-CLIP (arXiv:2401.06397) and CLOC (arXiv:2410.02746).
4. Lack of proofreading. It is very obvious that several articles in the References section are listed twice (e.g., arXiv:2505.05071, arXiv:2403.15378, arXiv:1306.5151, arXiv:2504.01916).

**Questions:**

1. Given that the method uses region-level/part-aware annotated data, and the design in lines 188-193 ("resize img -> CLS token") seems well-suited for bounding box classification (as opposed to ROIAlign or Attention Pooling), why was the model not evaluated on a box classification task?
2. The paper states, "This fundamental weakness is inherited by Multimodal Large Language Models (MLLMs) that use CLIP-based vision encoders," yet there are no experiments or analyses on MLLM-related benchmarks to support this or show improvement. Such an experiment could also address the "necessity of pretraining" mentioned in Weakness 2.

---

### Official Review · Reviewer_2jPT · 2025-10-31

**Soundness:** 2
**Presentation:** 2
**Contribution:** 1
**Rating:** 4
**Confidence:** 3

**Summary:**

This paper introduces PA-CLIP for FGVR and cross-modal retrieval. The method enhances CLIP in two main ways: first, by using a multi-level contrastive learning objective that incorporates both global and part-level descriptions; and second, by creating a new dataset, FG-Part, with ~1 million part-level image-text pairs for supervision. The model also extends CLIP's text capacity to 248 tokens and claims strong performance on 8 FGVR and 5 retrieval benchmarks.
Strengths

**Strengths:**

1、The paper correctly identifies a key limitation of CLIP-style models: their focus on global features makes them less sensitive to fine-grained, part-level differences. The proposed solution of adding part-level supervision is a logical and well-motivated approach to address this.
Simple and Practical Method. The approach is straightforward to implement. It adheres to the standard dual-encoder contrastive learning paradigm, simply adding new loss terms, which keeps the engineering overhead low.
3、The evaluation is extensive, spanning multiple FGVR and retrieval datasets. The ablation studies on different caption combinations also provide some useful insights into the source of the performance gains.

**Weaknesses:**

Substantial overlap with prior “long-text / fine-grained CLIP” work; stronger de-confounding is needed.
Your two pillars are long-text / multi-level global alignment and part-level alignment. But lifting CLIP’s 77-token limit and exploiting long descriptions has already been explored (e.g., Long-CLIP), and your pipeline (“extend text length, then add fine-grained supervision”) explicitly follows previous work. Yet there is no strong, like-for-like baseline against Long-CLIP / FG-CLIP, nor subtraction ablations showing how performance/generalization degrade without the long-text module or without the part module.

**Questions:**

1、Under the same data scale and compute budget, run end-to-end fair comparisons with Long-CLIP / FG-CLIP and report statistical significance; add subtraction ablations (remove long-text; remove part supervision) to establish necessity/sufficiency of each component.

2、In Related Work, more clearly position your method against Long-CLIP / FG-CLIP and articulate the incremental contribution.

3、Report FLOPs, peak memory, and latency for training/inference versus CLIP / SigLIP / Long-CLIP; clarify whether inference requires extra forward passes for multi-part crops.

4、Add hierarchy-aware evaluation on nearby subclasses—e.g., hierarchical consistency metrics and confusion matrices—to demonstrate that strong part alignment does not distort inter-class semantic topology.

---

### Official Review · Reviewer_4PL5 · 2025-11-01

**Soundness:** 2
**Presentation:** 2
**Contribution:** 2
**Rating:** 2
**Confidence:** 4

**Summary:**

The paper proposes Part-Aware CLIP (PA-CLIP), a method that improves fine-grained perception and interpretability in vision-language models by training them on part-level image-text pairs from a new dataset, FG-Part. The dataset consists of both existing human annotated object parts and synthetically created part annotation. A new part-level loss aligns the part-captions with a local crop of the object part. By aligning part-specific text to specific object regions, PA-CLIP enhances detail-aware recognition without sacrificing general performance.

**Strengths:**

- The paper addresses a well known limitation of CLIP by enabling more more fine-grained understanding of images.
- The methodology is simple, and tackles the problem mostly through additional supervised data.
- The results on fine-grained recognition tasks surpass those of baseline models such as FG-CLIP.

**Weaknesses:**

- The paper has very limited novelty.
  * The main technical change to prior approaches is the use of the part-level loss. However, it is just another CLIP loss trained of part-level annotations. Moreover, it is not clear why the part-level loss crops the object-region for the alignment. This approach discards contextual information that can contain useful and relevant information to identify the object part. An ablation should investigate whether this is the best solution.
  * The proposed training dataset builds upon existing data and enriches it with synthetic data. The quality of the data has not been validated. The scope of the data is rather narrow and focused on fine-grained object recognition.

- Some dataset details are missing.
  * It is not clear which images are sourced for the FG-Part dataset.
  * The number of total images in FG-Part dataset is not disclosed.

- Experimental evaluation is not convincing.
  * Tab. 1 shows clear improvements of PA-CLIP. However, the training dataset either includes these benchmark datasets or is specifically designed for them (e.g., CUB, StanfordCars, and FGVC-Aircraft). As a result PA-CLIP could be considered (partially) supervised on this task while the other methods are being evaluated zero-shot.
  * Tab. 2 results are unimpressive. PA-CLIP is roughly on par with FG-CLIP. The table likely does not contain SOTA baselines, e.g., FLAIR (which is referenced) is not included and seems to perform better for some of the retrieval tasks.
  * For evaluating a novel CLIP model, the experiments are not sufficient. Zero-shot ImageNet performance is a standard benchmark. Nowadays, models are typically evaluated in more breadth, e.g., zero-shot semantic segmentation, or as an encoder for MLLM training.
  * The interpretability results are not convincing. Very few qualitative examples are shown. There is no quantitative evaluation. The paper is motivated by direct embedding alignment so it is unclear why Grad-CAM is chosen instead of directly measuring cosine alignment with patches. It seems that the class prediction is explained instead of the part caption which would seem more natural.

**Questions:**

Please refer to the weaknesses section. Questions of particular interest are:
- What is the reason behind cropping the object-region for the part-level loss? How does it compare to processing the full image and then cropping at the patch level in feature space?
- Have you validated the synthetic data? How do you ensure its quality?
- Does the FG-Part dataset contain the training sets of all datasets in Tab. 1? Are there any other data sources?
- How many images are in the FG-Part dataset? How many annotations are there per image per dataset source?
- How does PA-CLIP perform on ImageNet?
- Can you show some visualizations of the patch alignment with text descriptions of parts?

---

### Meta-Review · Area_Chair_emVp · 2025-12-29

**Summary:**

This work studies the problem fine-grained visual recognition and cross-modal retrieval to improve the limited part-level understanding and interpretability of CLIP-style vision-language models. The motivation is the lack of explicit part-level supervision in existing vision-language pre-training. To tackle this challenge, this work proposed Part-Aware CLIP (PA-CLIP) that introduces multi-level contrastive learning and a new large-scale FG-Part dataset containing human-annotated and synthetic part-level image-text pairs. Experiments show strong performance of the proposed method on multiple fine-grained recognition benchmarks and competitive results on cross-modal retrieval tasks.

The main strengths are that (1) the research problem about fine-grained and part-level understanding for CLIP-style models is important, (2) the extension and implementation over CLIP is simple and practical, (3) the performance is strong on fine-grained recognition benchmarks (but the evaluation is questionable as mentioned in the following weaknesses).

The main weaknesses are that (1) the novelty is limited compared to prior work like FG-CLIP and Long-CLIP, (2) the misuse of training data leads to evaluation data leakage, causing an unfair zero-shot evaluation, (3) the key ablations on part-level loss are missing, (4) the evaluation on standard benchmarks with relevant recent methods are not comprehensive.

The authors did not respond to the reviewers' comments. Considering the overall evaluation of this paper, the final recommendation is reject.

**Reviewer Concerns:**

The authors did not provide a rebuttal, and the area chair shares the same major concerns with the reviewers.

**Reviewer Scores:**

The authors did not provide a rebuttal, and the scores would not be increased.

---

### Decision · Program_Chairs · 2026-01-26

Reject